# Autofluorescence imaging permits label-free cell type assignment and reveals the dynamic formation of airway secretory cell associated antigen passages (SAPs)

Viral S Shah[1,2†], Jue Hou[3†], Vladimir Vinarsky[2], Jiajie Xu[2], Manalee V Surve[2], Charles P Lin[3*], Jayaraj Rajagopal[1,2,4,5*]

[1]Department of Internal Medicine, Division of Pulmonary and Critical Care Medicine, Massachusetts General Hospital, Boston, United States; [2]Center for Regenerative Medicine, Massachusetts General Hospital, Boston, United States; [3]Advanced Microscopy Program, Center for Systems Biology and Wellman Center for Photomedicine, Massachusetts General Hospital, Harvard Medical School, Boston, United States; [4]Harvard Stem Cell Institute, Cambridge, United States; [5]Klarman Cell Observatory, Broad Institute, Cambridge, United States

*For correspondence:
charles_lin@hms.harvard.edu
(CPL);
jrajagopal@mgh.harvard.edu (JR)

[†]These authors contributed equally to this work

Competing interest: The authors declare that no competing interests exist.

**Abstract** The specific functional properties of a tissue are distributed amongst its component cell types. The various cells act coherently, as an ensemble, in order to execute a physiologic response. Modern approaches for identifying and dissecting novel physiologic mechanisms would benefit from an ability to identify specific cell types in live tissues that could then be imaged in real time. Current techniques require the use of fluorescent genetic reporters that are not only cumbersome, but which only allow the study of three or four cell types at a time. We report a non-invasive imaging modality that capitalizes on the endogenous autofluorescence signatures of the metabolic cofactors NAD(P) H and FAD. By marrying morphological characteristics with autofluorescence signatures, all seven of the airway epithelial cell types can be distinguished simultaneously in mouse tracheal explants in real time. Furthermore, we find that this methodology for direct cell type-specific identification avoids pitfalls associated with the use of ostensibly cell type-specific markers that are, in fact, altered by clinically relevant physiologic stimuli. Finally, we utilize this methodology to interrogate real-time physiology and identify dynamic secretory cell associated antigen passages (SAPs) that form in response to cholinergic stimulus. The identical process has been well documented in the intestine where the dynamic formation of SAPs and goblet cell associated antigen passages (GAPs) enable luminal antigen sampling. Airway secretory cells with SAPs are frequently juxtaposed to antigen presenting cells, suggesting that airway SAPs, like their intestinal counterparts, not only sample antigen but convey their cargo for immune cell processing.

## Editor's evaluation

The study addresses a significant impediment to deriving maximal value from living tissue culture systems, which is simultaneously distinguishing and mapping the activity of multiple constituent cell types over the course of experimental perturbations. The authors present a label-free approach that involves collecting autofluorescence and morphological features that provide distinct signatures for mouse tracheal epithelium and demonstrate its application for live imaging of secretory cells. This platform may be very valuable for answering specific experimental questions about tracheal cell behavior in disease.

**eLife digest** Imaging several cell types, at the same time, within a living tissue is no small endeavor. To do so, scientists usually have to perform genetic manipulations that make certain proteins in each cell type fluorescent and therefore easy to track. However, these approaches are cumbersome, limited, and often not applicable to intact human tissues.

A possible alternative would be to make use of autofluorescence – the fact that certain molecules in living cells naturally fluoresce when exposed to a particular wavelength of light. For example, this is the case for NAD(P)H and FAD, two small molecules necessary for life's biochemical processes, and whose intracellular levels and locations vary depending on cell type.

In response, Shah, Hou et al. developed a new imaging technique that takes advantage of the unique autofluorescence signatures of NAD(P)H and FAD to distinguish between the seven different types of cells that line the surface of the airways of mice.

Using their autofluorescence approach, Shah, Hou et al. were also able to discover a new role for secretory cells, which normally produce fluids, mucus and various proteins necessary for the lungs to work properly. The imaging experiments show that these cells could also sample material from the surface of the airway in a manner similar to how cells in the intestine take up material from the gut and pass their cargo to immune cells that mediate infection control or tolerance. Further studies should uncover more details about this new function of secretory lung cells. Other exciting discoveries may also emerge from researchers adopting the method developed by Shah, Hou et al. to examine a range of organs (both healthy and diseased), and attempting to apply it to human tissues.

## Introduction

The airway epithelium is composed of myriad distinct cell types, many of which have only been recently described (*Deprez et al., 2020*; *Goldfarbmuren et al., 2020*; *Montoro et al., 2018*; *Plasschaert et al., 2018*). Studying the function of each of these cells and their contribution to the physiology of the entire epithelial ensemble often requires cell type-specific genetic perturbation. Indeed, even the simple identification of a given class of cells within a live tissue requires genetic labeling with a fluorophore. Currently, cells are most commonly identified post-fixation using immunohistochemical and immunofluorescence staining, but this prohibits the assessment of real-time cellular dynamics and thus real-time physiology. Since post-fixation cell identification is destructive, it also precludes the study of long-term processes or the effect of multiple serial stimuli which are both essential features of normal organ physiology. Additionally, markers used for cell type identification in unperturbed tissue can often be non-specific or misleading after injury. We reasoned that a methodology for identifying and tracking diverse cell types within an organ in real time in a non-destructive fashion would permit the study of physiologically relevant processes that are currently hidden from view.

Label-free imaging based on endogenous fluorescence of metabolic cofactors has been deployed to measure cell metabolism in various tissues, particularly in tumors and immune cells (*Katz et al., 2002*; *Skala et al., 2007*). NADH, NADPH, and FAD, the major endogenous fluorophores, have been used to monitor metabolism in vitro and in vivo (*Gil et al., 2019*; *Heikal, 2010*; *Huang et al., 2002*; *Kasischke et al., 2004*; *Rocheleau et al., 2004*; *Sepehr et al., 2012*; *Tiede et al., 2007*). Since NADH and NADPH cannot be distinguished by their autofluorescence, the combination has been reported as an aggregate quantity, NAD(P)H (*Schaefer et al., 2019*). To account for the effects of light scattering and variation in the number of mitochondria, a metabolic ratio of FAD/(NAD[P]H+FAD) is often employed (*Chance et al., 1979*). Two-photon excited fluorescence microscopy can be used to measure this metabolic ratio in intact tissues and in vivo (*Dilipkumar et al., 2019*; *Klinger et al., 2012*; *Kreiß et al., 2020*). Indeed, abundant common intestinal epithelial cell types can be distinguished based on their morphology when imaged in this manner (*Kreiß et al., 2020*). Most recently, these measures of metabolic activity have been used to identify immune cells in differing functional states during fluorescence activated cell sorting (FACS) ex vivo (*Lemire et al., 2022*). Furthermore, immune cell interactions have been visualized using autofluorescence in the murine trachea (*Kretschmer et al., 2016*). In this manuscript, we describe a mouse tissue explant system that readily permits the label-free identification of abundant epithelial cell types in living airway tissue. Furthermore, a combined assessment of the cell resolution fine features of autofluorescence signatures can be coupled to cell morphometric

measurements to permit the identification of exceedingly rare epithelial cell types. We also use live autofluorescence imaging to reveal the existence of secretory cell associated antigen passages (SAPs) in the airway. The biology of such structures has only recently been dissected in the gut where they have been referred to as both SAPs and goblet cell associated antigen passages (GAPs) (*Gustafsson et al., 2021*; *Gustafsson and Johansson, 2022*; *Knoop et al., 2015*; *McDole et al., 2012*; *Noah et al., 2019*). We describe the real-time dynamics of the formation of these structures using physiologic cholinergic stimulation and their association with antigen-presenting cells (APCs). Finally, we demonstrate that autofluorescence signatures can be used for live cell tracing. Indeed, under some circumstances, this technique is more reliable than the use of ostensibly cell type-specific markers whose expression can be altered by injury, physiologic stimulus, and in disease states.

## Results

### Label-free autofluorescence-based live imaging of an intact murine tracheal explant

We hypothesized that we would be able to specifically assign airway epithelial cell type identity by directly imaging specific metabolic signatures at single-cell resolution. In order to preserve subtle and complex features of normal cellular and tissue architecture, we developed a mouse tracheal explant setup that allows upright two-photon imaging in a physiologic chamber over days (*Kwok et al., 2022*; *Figure 1A* and *Figure 1—figure supplement 1A*). To facilitate optimal detection and separation of NAD(P)H and FAD, the samples were scanned at 730 nm and 900 nm sequentially (*Huang et al., 2002*). A short pass 505 nm dichroic was used to separate the NAD(P)H and FAD signal, and each fluorescence signal was detected using 450/100 nm (NAD(P)H) and 540/80 nm (FAD) band pass filters, respectively (Supplement 1B). Using this setup, we ascertained NAD(P)H and FAD autofluorescence emissions across the excitation spectrum (*Figure 1B*). Due to the curvature of the trachea, we performed digital 'flattening' during image analysis where the subepithelial region was linearized to the same Z plane. To accomplish this, we utilized second harmonic generation (SHG) imaging of collagen fibers to identify the basement membrane (SHG signal was excited with a 900 nm laser) and developed a MATLAB algorithm to 'flatten' the epithelial surface (https://github.com/vss11/Label-free-autofluorescence, copy archived at *Shah, 2023*).

To verify that the measured autofluorescence signals reflect cellular metabolism in real time, we utilized pharmacologic agents to experimentally alter NAD(P)H and FAD. Treatment with rotenone and antimycin A, inhibitors of mitochondrial complex I and complex III, lead to a decrease in the FAD/(NAD[P]H+FAD) fluorescence ratio at the airway surface as predicted from similar experiments conducted in breast cancer cells (*Figure 1C*; *Hou et al., 2016*). Treatment with FCCP, a mitochondrial uncoupler, led to an anticipated increase in autofluorescence intensity ratio, as similarly previously documented (*Figure 1D*; *Hou et al., 2016*). In order to determine signal stability, we measured the autofluorescence intensity ratio over 2 days of tracheal explant culture. The signal remained stable (*Figure 1E*).

### Autofluorescence imaging discriminates the three common airway epithelial cell types and demarcates fine details of cell shape at the single-cell level

As previous studies have been able to visualize common cell types in the intestinal epithelium using autofluorescence (*Kreiß et al., 2020*), we first optimized our label-free imaging approach to identify the common airway epithelial cell types: ciliated cells, secretory cells, and basal cells. We first measured and recorded autofluorescence using live tissue and then subsequently identified cell types in that same tissue following fixation. Specifically, autofluorescence was first imaged in live tissues, locations were registered to subepithelial landmarks, and the same fixed tissue was subsequently stained with conventional markers to identify cell types. Live autofluorescence signatures were then associated with each cell that was identified by conventional staining. In order to register autofluorescence live images to that of fixed and stained images, we imaged the SHG signals of the subepithelial collagen. These signals served as fiduciary marks which facilitated exact localization of individual cells. We first visualized ciliated cells with live autofluorescence imaging given their very unique morphology. As seen in *Figure 2A*, ciliated cells have a robust FAD signal which is primarily located near the luminal

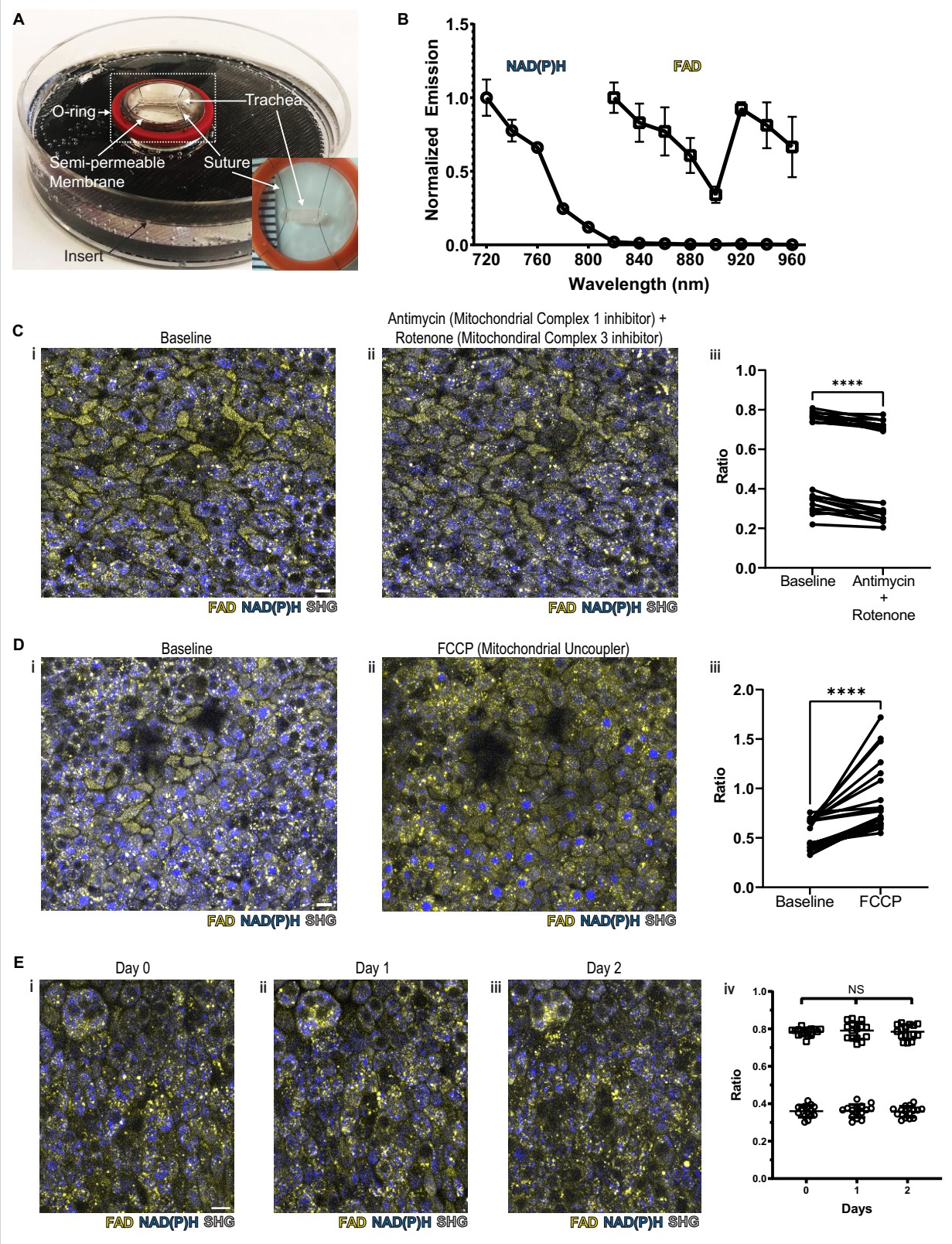

**Figure 1.** NAD(P)H and FAD autofluorescence can be measured from murine tracheal epithelial cells and is stable across days. (**A**) Mounting of tracheal explant for upright two-photon imaging. Semipermeable membranes (12 mm Transwell, 0.4 µm Pore Polyester Membrane Insert) were inverted and fixed to custom 3D printed inserts in 60 mm dishes using silicone adhesive. Trachea were dissected and sutured onto O-rings for additional stability (**A** inset and *Figure 1—figure supplement 1A*). Trachea and O-ring were then placed onto the semipermeable membrane for upright imaging and

*Figure 1 continued on next page*

*Figure 1 continued*

explant culture. Inset showing a trachea sutured to O-ring in a zoomed view. (**B**) Autofluorescence emission normalized to maximal emission at different excitation wavelengths for both NAD(P)H and FAD. (**C**) NAD(P)H (blue), FAD (yellow), and second harmonic generation (SHG, gray) autofluorescence images of murine tracheal explant at (**i**) baseline, (**ii**) with the addition of antimycin (10 µM – mitochondrial complex I inhibitor) and rotenone (1 µM – mitochondrial complex III inhibitor) showing increased NAD(P)H fluorescence, and (**iii**) quantification of autofluorescence ratio (FAD/FAD +NAD[P]H) demonstrating a decrease. Each point is a single cell. N=20. (**D**) NAD(P)H (blue), FAD (yellow), and SHG (gray) autofluorescence images of murine tracheal explant at (**i**) baseline, (**ii**) with the addition of carbonyl cyanide p-trifluoro-methoxyphenyl hydrazone (FCCP – mitochondrial uncoupler) showing decrease in NAD(P)H fluorescence and an increase in the metabolic ratio (**iii**). Each point is a single cell. N=20. (**E**) Time course images of NAD(P)H (blue), FAD (yellow), and SHG (gray) of the same region of a tracheal explant at day 0 (**i**), day 1 (**ii**), and day 2 (**iii**) showing similar fluorescence. (**iv**) Quantification of ratio (FAD/FAD +NAD[P]H) of individual cells showing stability across 0, 1, and 2 days. N=15. ****$p<0.0001$ by paired t-test. Scale bars = 10 µm.

The online version of this article includes the following figure supplement(s) for figure 1:

**Figure supplement 1.** Mouse tracheal explant and microscopy set up.

cell surface. Fixation and staining with acetylated tubulin (ac-tub, green) demonstrated coincidence of this autofluorescence signature with the presence of ciliated cells identified by immunofluorescence. Secretory cells demonstrated a more robust NADH signal (*Figure 2B*) that was in sharp contrast to that of ciliated cells. Post-fixation and staining with CCSP (magenta) demonstrates that the autofluorescence intensity ratio can also define secretory cell borders. Similarly, basal cells can also be identified by their autofluorescence signatures (*Figure 2C*) and correspond to cells positive for keratin 5 (KRT5, cyan). These differences in autofluorescence ratio can be quantified for the individual cell types and can be used to distinguish ciliated, secretory, and basal cells (*Figure 2D*).

As ciliated and secretory cells show the most well separated autofluorescence ratios, we assessed whether this difference would be maintained after pharmacologic intervention. Ciliated and secretory cells could still be distinguished following the application of the aforementioned mitochondrial poisons (*Figure 1C and D*), although the ratios were less well separated (*Figure 2—figure supplement 1*). This suggests that chemical or physiologic perturbations may alter the ability to distinguish cell types based on their autofluorescence ratios alone.

## Hillocks and rare airway epithelial cell types can be distinguished using a combination autofluorescence signatures coupled to simple metrics of morphology

We next assessed the ability of our protocol to identify hillocks, a newly identified airway epithelial structure (*Montoro et al., 2018*; *Figure 3A*). Due to the multilayered nature of hillocks, the structure is raised above the surrounding surface epithelia. The unique polygonal cell shape of hillock cells and discrete boundaries of hillocks are easily distinguishable with autofluorescence imaging. Post-fixation staining with keratin 13 (KRT13, magenta), a protein highly expressed by hillocks, demonstrates concordance with autofluorescence imaging.

We then took on the much greater challenge of identifying the very rare cell types of the epithelium: pulmonary ionocytes, tuft cells, and neuroendocrine cells (*Figure 3B–D*). Ionocytes (*Figure 3B*) had a characteristic shape by autofluorescence imaging which was similar to the shape of ionocytes delineated by immunohistochemical staining with BSND (green). Additionally, ionocytes were found to have puncta of high FAD signal toward the apical aspect of the cell (*Figure 3Biv*). The distinct shape of tuft cells could be identified by an autofluorescence ratio corresponding to immunofluorescent staining with GNAT3 (red). Of all airway epithelial cells, neuroendocrine cells labeled by PGP9 (cyan) displayed the lowest NAD(P)H and FAD autofluorescence (*Figure 3D*). Interestingly, as an aggregate, rare cells are characterized by very similar autofluorescence intensity ratios and could not be distinguished by this criterion alone (*Figure 3E*). Given the diversity of morphologies of airway cell types, we then quantified a number of morphometric measures to better distinguish rare cell identities (*Figure 3—figure supplement 1*). Live autofluorescence imaging permits the segmentation of cell boundaries enabling the measurement of the distance of the nucleus from the basement membrane (nuclear position), the largest dimension in the X, Y, and Z plane (X length, Y length, and Z length), the aspect ratios (X length/Y length) and (Y length/Z length), and the SD of fluorescence ratio intensities across the cell in the Z plane (SD of autofluorescence intensity ratio above the nucleus, SD of autofluorescence intensity ratio below nucleus, and the ratio of these two measures). We measured

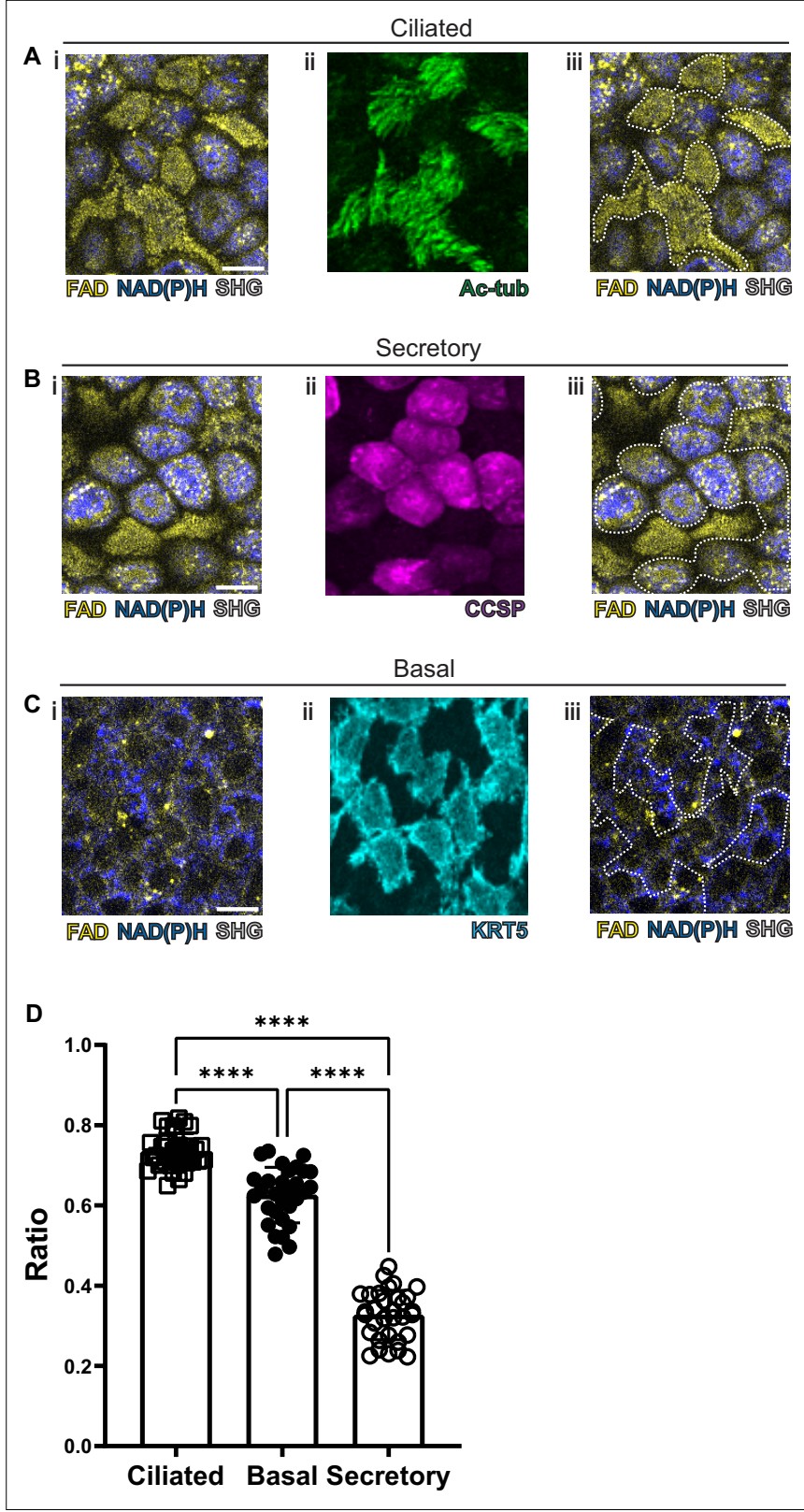

**Figure 2.** Autofluorescence imaging of common cell types. (**A–C**) NAD(P)H (blue), FAD (yellow), and second harmonic generation (SHG, gray) autofluorescence (**i**) coupled with cell type-specific staining (**ii**) with acetylated tubulin (**A**) (green) for ciliated cells (**B**) CCSP (magenta) for secretory cells and (**C**) KRT5 (cyan) for basal cells. (**iii**) Cell borders determined from post-fixation staining are demarcated on autofluorescence images.

*Figure 2 continued on next page*

*Figure 2 continued*

(**D**) Quantification of FAD/FAD +NAD(P)H for ciliated cells (N=32), secretory cells (N=32), and basal cells (N=30), demonstrating distinct autofluorescence ratio. ANOVA test for multiple comparisons. **** p<0.0001. Scale bars = 10 µm.

The online version of this article includes the following figure supplement(s) for figure 2:

**Figure supplement 1.** Autofluorescence ratios of ciliated cells and secretory cells after antimycin A + rotenone or FCCP application.

these parameters for the common cells (basal, ciliated, and secretory), hillocks, and rare cells (ionocyte, neuroendocrine, and tuft). Aggregated data from a total of 30 basal cells, 32 ciliated cells, 32 secretory cells, 30 hillock cells, 21 ionocytes, 30 tuft cells, and 31 neuroendocrine cells were compiled (*Figure 3E*).

Given the spread of various morphologic variables across different cell types, we performed unsupervised clustering of the cell types based upon both their autofluorescence intensity ratio and their morphological differences. In aggregate, these characteristics allowed the separation of all airway epithelial cell types into distinct clusters (*Figure 3F*, *Figure 3—figure supplement 2*). In order to assess the performance of machine learning algorithms designed to distinguish cell types, we divided our data set into training and testing subsets. We utilized 75% of the total cells (154 cells) for machine learning training, leaving 25% (52 cells) for subsequent validation. We then trained three machine learning models using k-nearest neighbors, multinomial logistic regression, and XGBoost algorithms to classify the different cell types. We then compared the identity assigned by the machine learning algorithms based on the autofluorescence and morphology data to the known identities determined by staining. K-nearest neighbor analysis and multinomial logistic regression resulted in 96% accuracy, while XGBoost performed with 92% accuracy. This was suggestive of high-quality modeling with a Matthews Correlation Coefficient of 0.95, 0.95, and 0.91 (*Figure 3—figure supplement 3*). We then established the weight that each characteristic contributes to cell type-specific identification when using a XGBoost machine learning algorithm (*Table 1*), which shows that nuclear position, aspect YZ, NADH autofluorescence, ratio of deviation, and FAD autofluorescence are the most important characteristics for cell type assignment. Upon excluding autofluorescence and using morphometric criteria alone, the accuracy and model quality as measured by a Matthews Correlation Coefficient are dramatically reduced, highlighting the importance of obtaining autofluorescence signatures for cell type identification (*Figure 3—figure supplement 4*).

## Autofluorescence imaging can be used to interrogate real-time physiology and demonstrates the existence of airway SAPs

In order to assess whether this methodology is useful to assess real-time physiology, we employed a paradigmatic physiologic stimulus and then assessed whether the tissue response could be monitored dynamically. Cholinergic and adrenergic stimulation have historically been known to cause airway secretory cell granule release (*Massaro et al., 1979*; *Yoneda, 1977*). The club cell secretory protein, CCSP (also known as CC10, Uteroglobin, CC16, and *SCGB1A1*), is a major constituent of these granules and is commonly used to assign secretory cell identity (*Wong et al., 2009*). We applied a classic cholinergic agonist, methacholine (which is used clinically to define airway hyper-responsiveness in asthma) to the airway explant model (*Adler et al., 2013*; *Fischer et al., 2019*; *Webber and Widdicombe, 1987*). Methacholine stimulation led to a 70% decrease in the CCSP immunofluorescence staining as measured by fluorescence intensity relative to time zero (*Figure 4A–B*). Indeed, this stimulus-induced reduction in CCSP, classically considered a specific cell type marker, leads to a dramatically diminished ability to identify secretory cells.

Methacholine stimulation also alters NAD(P)H and FAD fluorescence intensities; however, these autofluorescence intensity ratios continue to discriminate secretory cells (*Figure 4C–D*). Thus, we are able to assign secretory cell identity at the single-cell level even after stimulation, thereby allowing us to follow individual secretory cell physiologic responses over time.

To our surprise, we found that methacholine stimulation resulted in the development of non-fluorescent spherical 'voids' within secretory cells (*Figure 4C* boxes, further magnification of these boxes in *Figure 4E*). These 'voids' were distinct from a non-autofluorescent region that represented

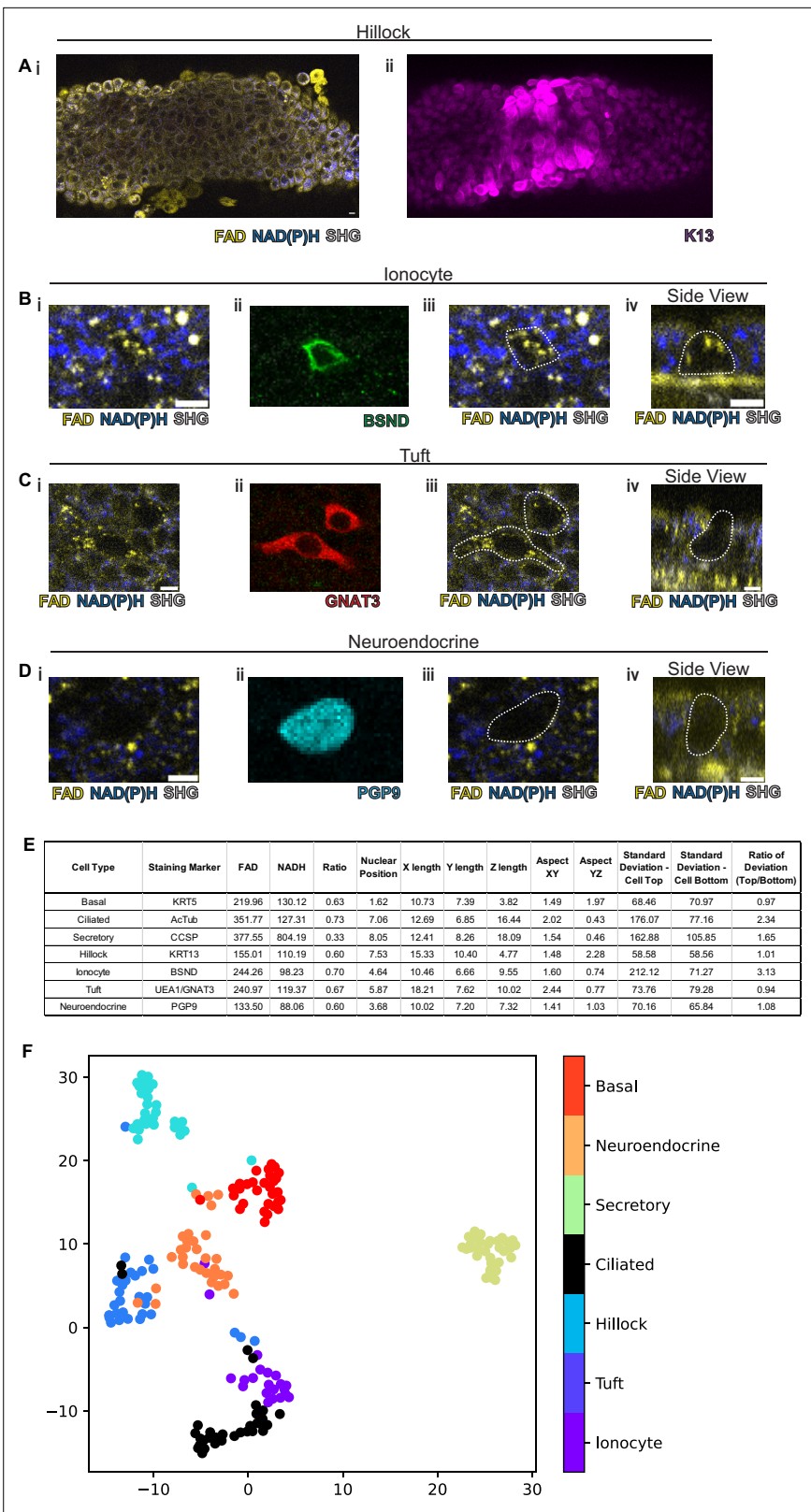

**Figure 3.** Rare cell detection using a combination of autofluorescence and unbiased clustering. NAD(P)H (blue), FAD (yellow), and second harmonic generation (SHG, gray) autofluorescence in (**i**) of (**A**) hillocks, (**B**) ionocytes, (**C**) tuft cells, and (**D**) neuroendocrine cells. Staining in (**ii**) of hillocks (K13 – magenta), ionocytes (BSND – green), tuft cells (GNAT3 – red), and neuroendocrine cells (PGP9 – cyan). Scale bars = 10 µm in (**A**) and 5 µm in (**B–D**). (**iii**)

*Figure 3 continued on next page*

*Figure 3 continued*

Overlay of cell outline from (**ii**) onto (**i**). Cross sectional imaging across the Z plane (**iv**) showing differential fluorescence specifically in ionocytes but absent in tuft and neuroendocrine (NE) cells. (**E**) Table with cell type-specific values of both autofluorescence (FAD, NADH, and ratio) and specific morphologic parameters. 'X-length'=largest length in the XY plane. 'Y-length'=smallest length in the XY plane. 'Z-length'=largest length in Z dimension. 'Aspect XY'=ratio of X-length/Y-length. 'Aspect YZ'=Y-length/Z-length. 'SD cell top'=SD of autofluorescence ratio from above mid height. 'SD cell bottom'=SD of autofluorescence ratio from below mid height. 'Ratio of SD' is SD cell top/SD cell bottom. (**F**) Unbiased clustering by UMAP from (**E**) demonstrating separate clusters for each cell type. The numbers of each cell type measured were as follows: ionocytes (21), tuft (30), hillock (30), ciliated (32), secretory (32), neuroendocrine (31), and basal (30).

The online version of this article includes the following source data and figure supplement(s) for figure 3:

**Source data 1.** Mean autofluorescence measurements (FAD, NADH, and ratio) and morphologic characteristics for various cell types as described in *Figure 3E*.

**Figure supplement 1.** Morphometric parameters used in the analysis of airway epithelial cell types.

**Figure supplement 2.** Unsupervised clusters analysis of all airway epithelial cell types shown individually.

**Figure supplement 3.** Confusion matrix demonstrating the performance of three algorithms for classifying cell identity.

**Figure supplement 4.** Confusion matrix demonstrating the performance of three algorithms, (**i**) k-nearest neighbors, (**ii**) multinomial logistic regression, and (**iii**) XGBoost, for classifying cell identity in the absence of NADH, FAD, and ratio autofluorescence data.

the nucleus. We performed detailed examination of the field in *Figure 4C* and found that 'voids' were present in approximately 9% of secretory cells prior to stimulation, but the fraction of secretory cells with 'voids' dramatically increased to 78% following methacholine stimulation (*Figure 4—source data 1*). We hypothesized that these novel structures resulted from the uptake of extracellular luminal contents. Indeed, such a sampling process has been demonstrated in the gut epithelium during which secretory and goblet cells internalize luminal contents. In this case, internalization results in the formation of SAPs and GAPs within the intestinal secretory and goblet cells (*Gustafsson and Johansson, 2022*; *McDole et al., 2012*; *Noah et al., 2019*). Given that airway secretory cells adopt a goblet cell morphology within hours, airway SAPs and GAPs are likely equivalent (*Evans et al., 2004*).

To definitely evaluate our hypothesis concerning the origin of the spherical autofluorescence 'voids' in our system, we incubated tracheal explants cultures in 10Kd FITC-dextran. Upon stimulation with methacholine, secretory cells showed uptake of FITC-dextran that was clearly distinguishable from the nucleus (*Figure 5A*). Since autofluorescence imaging alone cannot be used to establish the presence of a membrane, we resorted to employing epithelia expressing membrane tdTomato to assess whether the 'voids' were membrane bound. Indeed, we found uptake of FITC-dextran into membrane encapsulated structures delineated by membrane tdTomato (*Figure 5B*). This uptake of dextran occurs both in unstimulated and methacholine stimulated samples, though is dramatically exaggerated following stimulation with methacholine (*Figure 5C*). The increased SAP formation following stimulation with methacholine recapitulates the induction of intestinal GAPs following cholinergic stimulation (*Knoop et al., 2015*). In order to visualize this process on a faster time scale, we utilized air liquid interface (ALI) cultures since they are significantly thinner than the native trachea, thus allowing rapid imaging across the Z-axis. In our ALI models, we captured the fine dynamics of the formation of intracellular membrane bound structures that housed contents that were previously located in the airway lumen. Live imaging over 60 min demonstrates the uptake of FITC-dextran into the cell from the apical surface, further accumulation of FITC-dextran into the cell, and a major secretion event

**Table 1.** Contribution of each autofluorescence or morphometric characteristic to cell type-specific identification using XGBoost machine learning algorithm.

| FAD | NADH | Ratio | Nuclear position | X length | Y length | Z length | Aspect XY | Aspect YZ | SD - cell top | SD - cell bottom | Ratio of deviation (top/bottom) | Total |
|---|---|---|---|---|---|---|---|---|---|---|---|---|
| 0.10 | 0.12 | 0.06 | 0.18 | 0.09 | 0.01 | 0.08 | 0.02 | 0.13 | 0.07 | 0.02 | 0.11 | **1.00** |

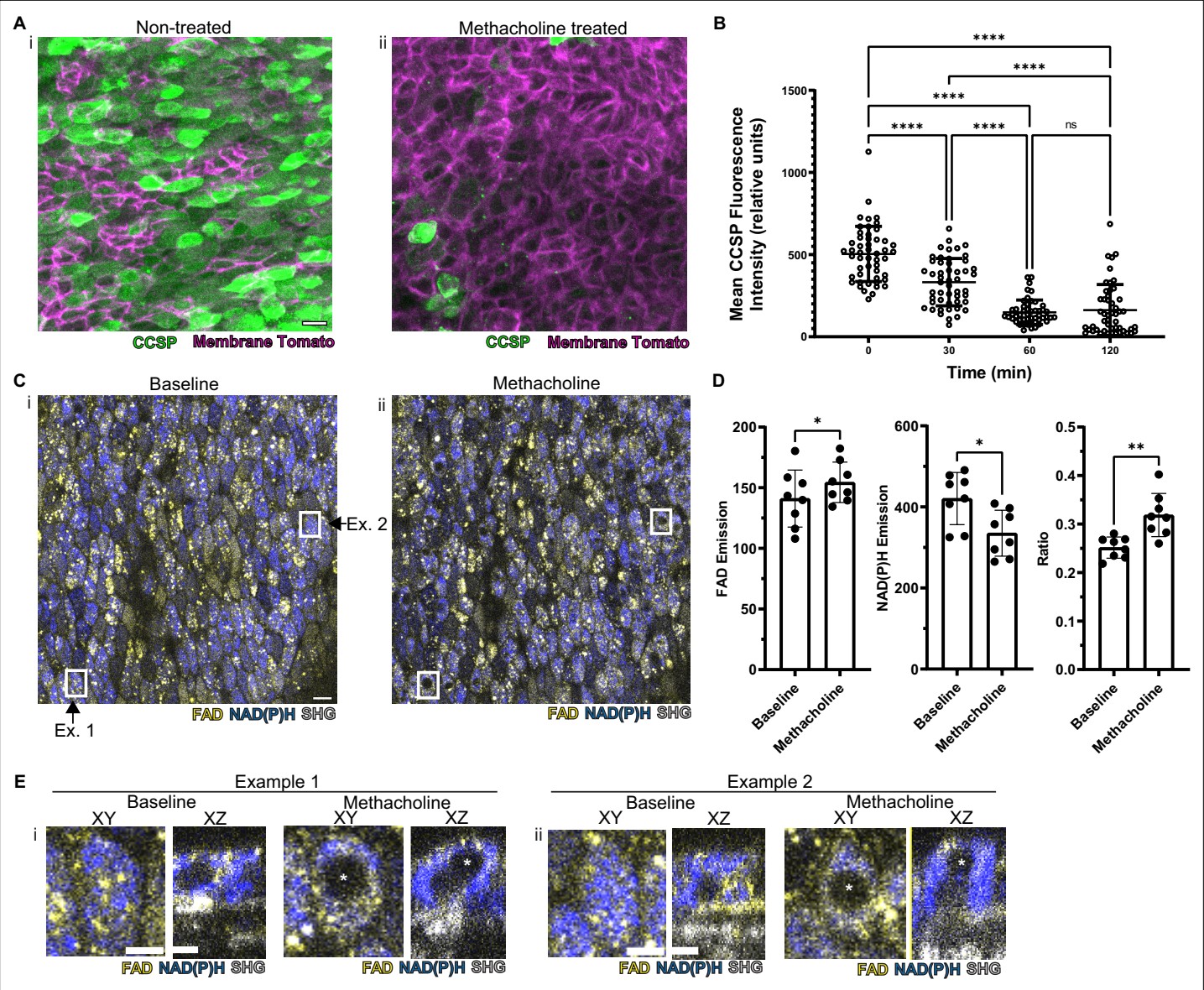

**Figure 4.** Autofluorescence imaging reliably identifies secretory cells despite the loss of characteristic secretory cell CCSP staining following methacholine stimulation. (**A**) Murine airway epithelial cells expressing membrane tomato under ROSA promoter. CCSP staining (green) using identical imaging parameters performed on two adjacent halves of same trachea treated with (**i**) vehicle vs (**ii**) 10 μM methacholine. CCSP staining (green) is lost following methacholine stimulation. Cell membrane denoted by membrane tomato (magenta). (**B**) Fluorescence intensity of CCSP staining over time with methacholine stimulation imaged with same parameters. Each cell is represented by a dot. N=50. ****p<0.0001 by ANOVA. (**C**) Autofluorescence imaging (NADH blue, FAD yellow, and second harmonic generation [SHG] gray) of trachea at baseline (**i**) and after methacholine stimulation (**ii**). Magnified image of the region indicated by the white box is shown in (**E**). (**D**) Quantification of NADH, FAD, and fluorescence ratio (FAD/FAD +NAD[P]H) at baseline and after methacholine stimulation at single-cell resolution (each dot represents a cell, N=8). (**E**) White boxed regions from (**C**) showing the formation of non-fluorescent voids after methacholine stimulation in both an XY view and a cross-sectional XZ view. Non-fluorescent regions in baseline XZ are nucleus. Asterix denotes new non-fluorescent 'void' after methacholine stimulation. Scale bars = 10 μm in A and C and 5 μm in E.

The online version of this article includes the following source data for figure 4:

**Source data 1.** Secretory cells with "voids" are present at baseline and their numbers increase following 30 minutes of methacholine treatment.

(*Figure 5D* and *Figure 5—videos 1–3*, *Figure 5—figure supplement 1*). This process bears a striking resemblance to the formation of intestinal SAPs and GAPs which are generated through an endocytic process that can be stimulated by acetylcholine (*Gustafsson et al., 2021*; *Knoop et al., 2015*). Additionally, further mirroring their gut counterparts, airway secretory cells with SAPs containing either FITC-dextran or FITC-ovalbumin closely associate with CD11c+APCs that had been recruited

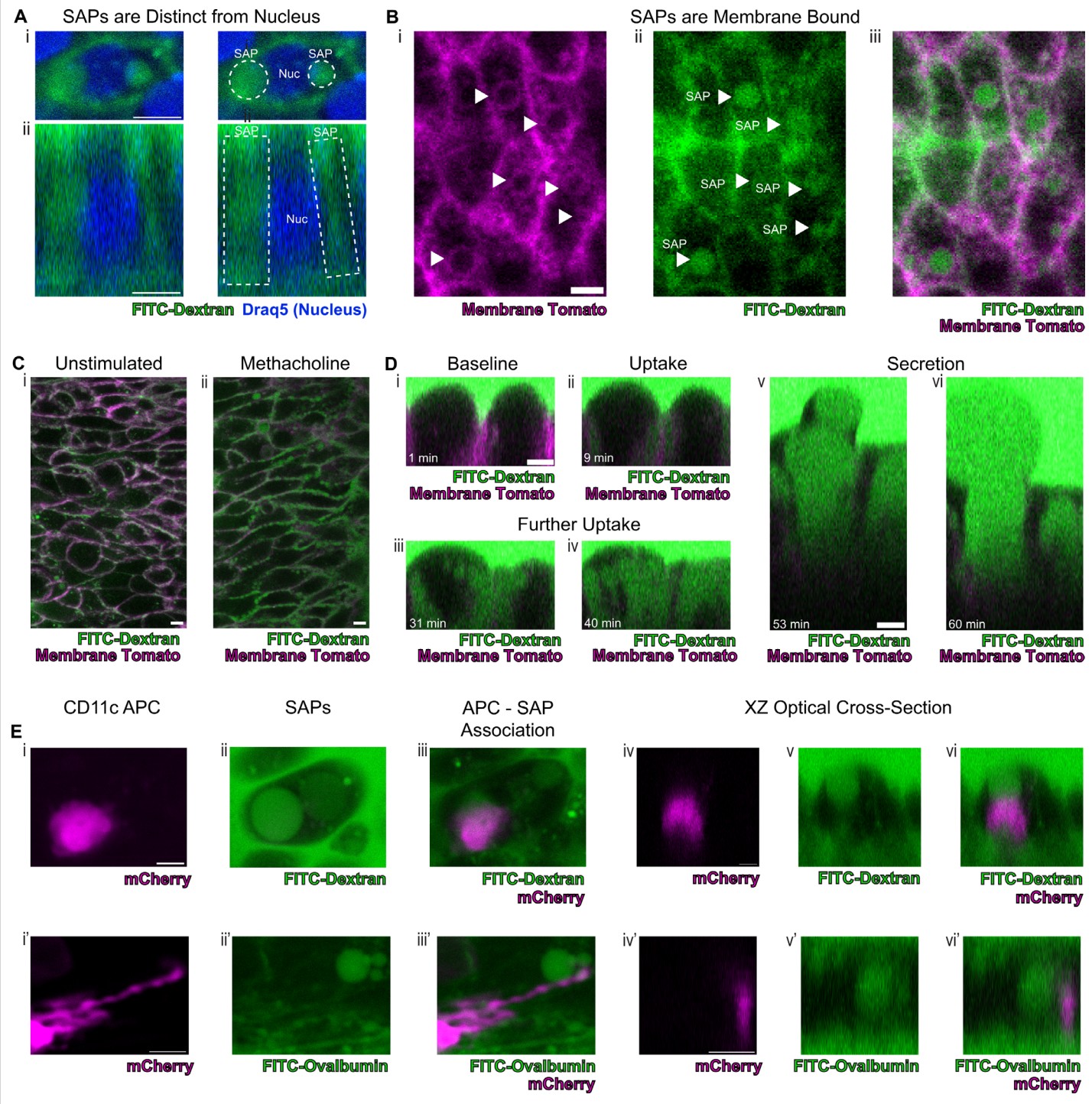

**Figure 5.** Secretory cells take up luminal contents and form secretory cell associated antigen passages (SAPs). (**A**) Secretory cell demonstrating uptake of FITC-dextran (10Kd, green) that is distinct from the nucleus stained by Draq5 (magenta). (**i**) Top panels are xy views and (**ii**) bottom panels are cross-sectional xz view. Dotted lines outline SAPs. Scale bars = 5 µm. (**B**) Tracheal imaging of (**i**) membrane tomato (magenta) and (**ii**) FITC-dextran (**iii**) overlay of both channels demonstrates methacholine-stimulated uptake of extracellular dextran into membrane bound SAPs (arrowheads). Scale bars = 5 µm. (**C**) FITC-dextran uptake in unstimulated and methacholine stimulated tracheal explants. Trachea from a mouse expressing membrane tomato (magenta) was bisected, and one half of the trachea was placed in media containing FITC-dextran (green) in the absence of methacholine, while the other half was placed in the same FITC-dextran media containing 10 µM methacholine. After 30 min of incubation, dextran uptake occurs in (**i**) unstimulated samples and (**ii**) dramatically increases following methacholine stimulation. (**D**) Selected images from **Figure 5—video 1** demonstrating uptake of luminal FITC-dextran and subsequent secretion of cell contents. (**i**) Baseline image prior to methacholine stimulation. (**ii**) Uptake of FITC-dextran and formation of

*Figure 5 continued on next page*

*Figure 5 continued*

SAPs (9 min after methacholine stimulation). (**iii** and **iv**) Further accumulation of FITC-dextran (31 min and 40 min after methacholine stimulation). (**v and vi**) Secretion of cellular contents (53 min and 60 min after methacholine stimulation). (**E**) CD11c+antigen-presenting cells (APCs) interact with SAPs. (**i** and **i'**) CD11c+APCs labeled by mCherry (magenta) (**ii** and **ii'**) methacholine stimulated SAP formation with FITC-dextran uptake (**ii**) and FITC-ovalbumin uptake (**ii'**). (**iii** and **iii'**) CD11c+APCs associate with SAPs (**iv–vi** and **iv'–vi'**) optical XZ cross sections of CD11c+APCs juxtaposed to SAPs. Scale bars = 5 μm.

The online version of this article includes the following video and figure supplement(s) for figure 5:

**Figure 5—video 1.** Movies of murine airway liquid interface (ALI) cultures using airway epithelial cells labeled with membrane tomato (magenta) demonstrating dextran (green) uptake and secretion.

https://elifesciences.org/articles/84375/figures#fig5video1

**Figure 5—video 2.** Movies of murine airway liquid interface (ALI) cultures using airway epithelial cells labeled with membrane tomato (magenta) demonstrating dextran (green) uptake and secretion.

https://elifesciences.org/articles/84375/figures#fig5video2

**Figure 5—video 3.** Movies of murine airway liquid interface (ALI) cultures using airway epithelial cells labeled with membrane tomato (magenta) demonstrating dextran (green) uptake and secretion.

https://elifesciences.org/articles/84375/figures#fig5video3

**Figure supplement 1.** XZ optical cross-sectional view of a murine airway liquid interface culture (ALI) of airway epithelial cells expressing membrane tomato (magenta) cultured in media containing FITC-dextran (green) demonstrating numerous examples of dextran uptake and secretion.

**Figure supplement 2.** CD11c+antigen-presenting cells (APC) associate with secretory cell associated antigen passages (SAPs).

to the airway epithelium following an LPS challenge (*Figure 5E*, *Figure 5—figure supplement 2*; *McDole et al., 2012*). Many of these LPS-stimulated APCs are intraepithelial and possess dendriform processes that extend toward SAPs. This suggests that airway SAPs are likely to play an important role in the immunologic response to antigens present in the airway lumen.

## Discussion

We report a label-free imaging methodology that can be used for cell type identification in live and unmanipulated tissues. We also show that autofluorescence coupled to morphology allows one to distinguish all the known airway epithelial cell types without the need for genetic labels. Unexpectedly, cell identity can be established in a way that is sometimes more accurate than destructive immuno-fluorescent approaches. This may be of particular relevance to disease states since it has been noted that markers of cell type such as the secretory cell marker CCSP can be misleading when used for the enumeration of cell numbers in COPD patient samples (*Pilette et al., 2001*), in asthma (*Shijubo et al., 1999*), and in smokers (*Shijubo et al., 1997*) presumably due to the fact that CCSP has been secreted into the lumen. However, despite its advantages, it should be noted that infection and inflammation may alter autofluorescence (*Dilipkumar et al., 2019*). Thus, our methodology must be re-optimized in each new experimental setting, and in each case, it is very likely that machine learning algorithms will require re-training. However, the training set provided in this manuscript will be a useful validation for native mouse tracheal epithelia. With these caveats in mind, we hope our methodology will eventually find applications in assessing human airway specimens, including the real-time assessment of disease-specific physiologic parameters and the efficacy of therapeutic interventions.

Finally, our discovery of airway SAPs demonstrates that this new methodology can be used to define novel unappreciated physiologic processes that currently remain hidden from view. Recently, using a new fixation procedure, luminal dextran uptake has been documented in murine airway goblet and secretory cells, suggestive of GAPs (*Tang et al., 2022*). We now definitively demonstrate endocytic uptake of luminal antigens, and we show that this process is stimulated by a classic cholinergic challenge (*Gustafsson and Johansson, 2022*; *McDole et al., 2012*). However, although we see that SAPs are associated with APCs, we have not demonstrated that APCs receive cargo from SAPs.

Thus, the actual functions of airway SAPs are yet to be discovered. Intestinal GAPs play a role in immune tolerance (*Knoop et al., 2017*; *Knoop et al., 2020*), the imprinting of dendritic cells (*Kulkarni et al., 2020*), and participate in inflammatory disease (*Knoop et al., 2017*; *Knoop et al., 2020*). Analogous phenomenon has been documented to occur in the case of goblet cells of the eye (*Barbosa et al., 2017*; *Ko et al., 2018*). It will be of great interest to establish the functions of airway SAPs and their role in airway mucosal immunity, tolerance, and autoimmunity.

# Materials and methods

**Key resources table**

| Reagent type (species) or resource | Designation | Source or reference | Identifiers | Additional information |
|---|---|---|---|---|
| Antibody | Mouse monoclonal anti-tubulin, acetylated | Sigma | T-6793 | 1:500 |
| Antibody | Rabbit monoclonal anti-BSND | Abcam | ab196017 | 1:150 |
| Antibody | Goat polyclonal anti-GNAT3 | Novus | NBP1-20926 | 1:100 |
| Antibody | Rabbit monoclonal anti-KRT13 | Abcam | ab92551 | 1:100 |
| Antibody | Goat polyclonal anti-CC10 | Santa Cruz | SC-9772 | 1:1000 |
| Antibody | Rabbit polyclonal anti-UCHL1 | Proteintech | 14730–1-AP | 1:250 |
| Antibody | Chicken polyclonal anti-KRT5 | Biolegend | 905901 | 1:200 |
| Chemical compound and drug | Acetyl-β-methylcholine chloride | Sigma | A2251 | |
| Chemical compound and drug | 10Kd FITC-dextran | Sigma | FD10S | |
| Chemical compound and drug | FITC-ovalbumin | Thermo | O23020 | |
| Chemical compound and drug | ROCK inhibitor Y-27632 | Selleckbio | S1049 | |
| Chemical compound and drug | Antimycin A from Streptomyces sp | Sigma | A8674 | |
| Chemical compound and drug | Rotenone | Sigma | 557368 | |
| Chemical compound and drug | Carbonyl cyanide 4-(trifluoromethoxy) phenylhydrazone | Sigma | C2920 | |
| Strain and strain background (*Mus musculus*) | C57BL/6J | Jax | stock no. 000664 | |
| Strain and strain background (*Mus musculus*) | MT-MG | Jax | stock no. 007676 | |
| Strain and strain background (*Mus musculus*) | Cd11c-mCherry | *Khanna et al., 2010* | | |
| Software and algorithm | MATLAB | Mathworks | https://www.mathworks.com/products/matlab.html | |
| Software and algorithm | Jupyter Notebook | Jupyter | https://jupyter.org/ | |

*Continued on next page*

*Continued*

| Reagent type (species) or resource | Designation | Source or reference | Identifiers | Additional information |
|---|---|---|---|---|
| Software and algorithm | FIJI | *Schindelin et al., 2012* | https://fiji.sc/ | |
| Software and algorithm | GraphPad Prism | GraphPad | https://www.graphpad.com | |
| Other | 12 mm Transwell | Corning | 3460 | Semipermeable membrane support |
| Other | Sylgard 184 silicone elastomer | Electron Microscopy Sciences | EMS Catalog #24236–10 | Silicone adhesive used for fixation of transwell insert |
| Other | Silicone O-ring | McMaster-Carr | 1182N015 | O-ring used to secure trachea |
| Other | 3D printed insert | This manuscript | https://github.com/vss11/Label-free-autofluorescence | 3D printed insert used to secure semipermeable membrane |
| Other | 7–0 Silk Suture | Ethicon | 786 G | Suture used to secure trachea to O-ring |
| Other | UEA1-dylight 649 | Vector Laboratories | DL-1068–1 | 1:100, lectin stain |
| Other | Draq5 | Abcam | ab108410 | 1:5000, nuclear stain |

## Mouse models

C57BL/6J mice (stock no. 000664) and MT-MG (stock no. 007676) were purchased from Jackson laboratory. Cd11c-mCherry mice were a gift from the Khanna lab (*Khanna et al., 2010*).

Mice were maintained in accordance with the Association for Assessment and Accreditation of Laboratory Animal Care-accredited animal facility at the Massachusetts General Hospital (MGH). All procedures were performed with Institutional Animal Care and Use Committee (IACUC)-approved protocols. All mice were housed in an environment with controlled temperature and humidity, on 12 hr light:dark cycles, and fed with regular rodent's chow.

## Mouse tracheal explant model

As demonstrated in *Figure 1—figure supplement 1A*, mice were euthanized as per MGH IACUC protocols, and trachea were dissected, cleared of connective tissue, opened longitudinally, and bisected. The dissected trachea was then sutured onto a silicone O-ring and placed onto a custom-made insert. The custom made insert was assembled using an inverted 12 mm Transwell, 0.4 µm Pore Polyester Membrane Insert fixed to a custom 3D printed insert (https://github.com/vss11/Label-free-autofluorescence) with Sylgard 184 silicone elastomer (EMS Catalog #24236–10).

Samples were incubated with Gibco DMEM/F12 (Fischer 11320033). For experiments across multiple days, the insert was filled with media such that the meniscus was located below the trachea to prevent explant submersion. Primocin (InVivoGen) was added to the media to prevent contamination.

## Two-photon microscopy

Tracheal explants were fixed to inserts as described above. An Olympus FVMPE-RS multiphoton laser scanning microscope was equipped with a MaiTai HPDS-O IR pulsed laser (900 nm for FAD and SHG), an INSIGHT X3-OL IR pulsed laser (730 nm for NAD[P]H), and a 25× water immersion lens (NA 1.05; XLPN25SWMP2) for imaging. Media was used as the submersion droplet. Configuration of the microscope setup is seen in *Figure 1—figure supplement 1B*. A 2× optical zoom with 1024×1024 scan

size was used. A full thickness Z section was imaged from above the tracheal surface through to the sub epithelium, with a 0.5 µm step size. Multiple regions on the same tracheal explant were imaged. Each region was 'registered' using SHG features of the subepithelial collagen/cartilage/vasculature as fiduciary marks. The insert was placed in a physiological live imaging chamber ($CO_2$ and temperature-controlled, TokaiHit) at 37°C and 5% $CO_2$.

## Mitochondrial poisons

Tracheal explants were incubated with DMEM/F-12 Media with Primocin (InVivoGen) and 15 mM HEPES. After baseline imaging, samples were incubated with complex 1 inhibitor – antimycin A (10 µM) and complex III inhibitor – rotenone (1 µM) for 10 min. Imaging was conducted as described. Similarly, after baseline imaging, tracheal explants were incubated with media containing mitochondrial uncoupler, carbonyl cyanide p-trifluoro-methoxyphenyl hydrazone (FCCP, 1 µM).

## Immunofluorescence staining of tracheal explants

Tracheal explants were secured onto the insert setup as described above. Tracheas were rinsed in PBS and subsequently fixed in 4% PFA in PBS for 20 min at room temperature. A glass slide was placed a few millimeters above the tissue to form a column of liquid. Explants were subsequently washed with PBS three times for a total of 1 hr. The explants were then permeabilized in PBS-0.3% TritonX-100 (PBST) for 30 min in a similar manner. The sample was then stained with primary antibody at 37°C for 1 hr, diluted in 1% BSA-0.3% PBST. The following antibodies and dilutions were used:

| Antibody | Manufacturer | Dilution |
| --- | --- | --- |
| Mouse anti-tubulin, acetylated | Sigma T-6793 | 1:500 |
| Rabbit anti-BSND | Abcam ab196017 | 1:150 |
| Goat anti-GNAT3 | Novus NBP1-20926 | 1:100 |
| Rabbit anti-KRT13 | Abcam ab92551 | 1:100 |
| UEA1-dylight 649 | Vector Laboratories DL-1068–1 | 1:100 |
| Goat anti-CC10 | Santa Cruz SC-9772 | 1:1000 |
| Rabbit anti-UCHL1 | Proteintech 14730–1-AP | 1:250 |
| Chicken anti-KRT5 | Biolegend 905901 | 1:200 |
| Draq5 | Abcam ab108410 | 1:5000 |

Tracheal explants were then washed with PBS for 1 hr and stained with secondary antibody at 37°C for 1 hr. Various secondary antibodies (Jackson ImmunoResearch) were used. The samples were then imaged using two-photon microscopy as above.

## Cell autofluorescence and morphological analysis

Autofluorescence images were obtained as described above. In order to normalize the Z-axis to the basement membrane (SHG), or 'flatten' the image, a MATLAB algorithm was utilized. Autofluorescence intensity was measured at the epithelial surface in the NAD(P)H and FAD channels using Image J. The morphological measurements were conducted as described in *Figure 3—figure supplement 1*.

## Dimensionality reduction and visualization

UMAP was calculated using the package in Python with the following parameters: random state = 42; spread = 3; min; dist = 0.1. A total of 206 cells were analyzed. Data visualization was implemented using the Matplotlib package in Python. All codes are provided in the supplemental Jupyter notebook file.

## Machine learning algorithms for classifying cell types

K-nearest neighbor, multinomial logistic regression, and XGBoost algorithms were used to classify cell types. The Scikit-learn package was used to implement k-nearest neighbor and multinomial logistic regression algorithms, and the XGBoost package was used to implement the XGBoost algorithm.

All the codes are provided in the supplemental Jupyter notebook file. The following features were used to train the algorithms: 1. FAD; 2. NADH; 3. FAD/NADH ratio; 4. nuclear position; 5. X-length; 6. Y-length; 7. Z-length; 8. XY aspect ratio; 9. YZ aspect ratio; 10. SD of the fluorescence ratio from midpoint above (cell top); 11. SD of fluorescence ratio from midpoint below (cell bottom); 12. SD of the fluorescence ratio top/SD of fluorescence ratio bottom. 152 cells (75%) were used to train the model, and 52 cells (25%) were used to test the models. For k-nearest neighbor algorithm, k=3 yields the highest accuracy and was therefore used to generate the confusion matrix. The Matthew's Correlation Coefficient was calculated using the Scikit-learn package.

## Methacholine treatment and CCSP immunofluorescence staining

Tracheal explants from mice expressing membrane tomato under the ROSA promoter (stock no. 007676) were euthanized and dissected as described above. Tracheal samples were bisected as described with one half placed in DMEM/F12 media, while the other half was placed into DMEM/F12 with 10 µM of methacholine (dry powder dissolved in DMEM/F12, Sigma A2251). After 30 min, 1 hr, 1.5 hr, and 2 hr of incubation, the samples were fixed in 4% PFA in PBS for 20 min at room temperature with agitation. Samples were then washed with PBS three times for a total of 1 hr. Samples were permeabilized in PBS-0.3% Triton X-100 (PBST) for 30 min with gentle agitation. The samples were stained with CC10 primary antibody (aka Scgb1a1, 1:500; SC-9772, Santa Cruz) at 37°C for 2 hr, diluted in 1% BSA-0.3% PBST. Samples were rinsed in PBS for 30 min and incubated with secondary antibody for 30 min with a subsequent 30 min PBS wash. As above, imaging was conducted on the Olympus FVMPE-RS multiphoton laser scanning microscope equipped with a MaiTai HPDS-O IR pulsed laser (900 nm for AF488) and an INSIGHT X3-OL IR pulsed laser (1050 nm for membrane tomato). Laser power was kept constant for all images, and all tracheal samples were acquired using the same settings. ImageJ was used for quantifying fluorescence intensity on maximal projection images.

## FITC-dextran and FITC-ovalbumin uptake assays

Explants were incubated with 10Kd FITC-dextran (Sigma FD10S) diluted into DMEM/F12 media at a concentration of 1 mg/ml with 10 µM methacholine (dry powder dissolved in DMEM/F12, Sigma A2251). Alternatively, explants were treated with FITC-ovalbumin (Thermo O23020) diluted into DMEM/F12 media at a concentration of 0.1 mg/ml with 10 µM methacholine (dry powder dissolved in DMEM/F12, Sigma A2251). Imaging was conducted on the Olympus FVMPE-RS multiphoton laser scanning microscope equipped with a MaiTai HPDS-O IR pulsed laser (800 nm for FITC) and an INSIGHT X3-OL IR pulsed laser (1050 nm for Draq5 or membrane tomato) with 2× optical zoom, 1024×1024 scan size, and a step size of 0.5 µm.

## APC colocalization with SAPs

Cd11c-mCherry mice were stimulated with LPS (30 µg in 40 µL PBS, delivered intranasally as a single dose, 5 days prior to experimentation). Explants from these mice were then cultured in DMEM/F12 containing 1 mg/ml 10Kd FITC-dextran (Sigma FD10S) and 10 µM of methacholine (Sigma A2251). Imaging was conducted on the Olympus FVMPE-RS multiphoton laser scanning microscope equipped with a MaiTai HPDS-O IR pulsed laser (800 nm for FITC) and INSIGHT X3-OL IR pulsed laser (1050 nm for mCherry) with 1024×1024 scan size and a step size 0.5 µm. A 15× optical zoom and line averaging scan were used for high-resolution imaging.

## Air liquid interface model

Tracheal cells were harvested and cultured as previously described (*Mou et al., 2016*). Briefly, cells were cultured and expanded in complete SAGM (small airway epithelial cell growth medium; Lonza, CC-3118) containing a TGF-β/BMP4/WNT antagonist cocktail and 5 µM ROCK inhibitor Y-27632 (Selleckbio, S1049). To generate ALI cultures, airway basal stem cells were seeded onto transwell membranes and differentiated using PneumaCult-ALI Medium (StemCell, 05001). Air-liquid cultures were fully differentiated after 14 days as demonstrated by the formation of air-liquid interface and the beating of cilia.

## Statistical analysis

All analysis was performed using Graphpad Prism. For analysis of experimental data after treatment with mitochondrial poisons (*Figure 1C–D*), imaging over time (*Figure 1E*), and methacholine

treatment (*Figure 4D*), a paired t-test was used. To assess change in autofluorescence after metabolic poisons (*Figure 2—figure supplement 1*), an unpaired t-test was used. For comparison of autofluorescence ratio across different cell types (*Figure 2D*), one-way ANOVA was utilized. For comparison of immunofluorescence staining with various durations of methacholine treatment (*Figure 4B*), a one-way ANOVA was used.

## Acknowledgements

We would like to thank the members of the Rajagopal lab and Lin labs for their feedback. This work was supported by NIH-NHLBI 5R01HL142559-04 (JR, CPL), RO1HL118185-08 (JR), 1R01HL157221-01A1 (JR), the Bernard and Mildred Kayden Endowed MGH Research Institute Chair (JR), and CFF 003338L121 (VS).

## Additional information

### Funding

| Funder | Grant reference number | Author |
|---|---|---|
| Cystic Fibrosis Foundation | 003338L121 | Viral S Shah |
| NHLBI Division of Intramural Research | 5R01HL142559 | Charles P Lin Jayaraj Rajagopal |
| NHLBI Division of Intramural Research | RO1HL118185 | Jayaraj Rajagopal |
| NHLBI Division of Intramural Research | 1R01HL157221 | Jayaraj Rajagopal |

The funders had no role in study design, data collection and interpretation, or the decision to submit the work for publication.

### Author contributions

Viral S Shah, Conceptualization, Formal analysis, Investigation, Visualization, Methodology, Writing - original draft, Writing – review and editing; Jue Hou, Conceptualization, Formal analysis, Investigation, Visualization, Methodology, Writing – review and editing; Vladimir Vinarsky, Conceptualization, Investigation, Methodology; Jiajie Xu, Manalee V Surve, Investigation, Methodology, Writing – review and editing; Charles P Lin, Jayaraj Rajagopal, Conceptualization, Resources, Formal analysis, Supervision, Funding acquisition, Investigation, Methodology, Writing – review and editing

### Author ORCIDs

Viral S Shah http://orcid.org/0000-0002-3605-8066
Vladimir Vinarsky http://orcid.org/0000-0003-1141-6434
Jayaraj Rajagopal http://orcid.org/0000-0002-4122-177X

### Ethics

Mice were maintained in accordance with the Association for Assessment and Accreditation of Laboratory Animal Care-accredited animal facility at the Massachusetts General Hospital (MGH). All procedures were performed with Institutional Animal Care and Use Committee (IACUC)-approved protocols (#2009N000119). MGH is accredited by AAALAC International, has an assurance with the Office of Laboratory Animal Welfare (OLAW) and is registered with the United States Department of Agriculture (USDA). Euthanasia was performed via house line $CO_2$-mediated asphyxiation and confirmatory cervical dislocation consistent with the recommendations of the American Veterinary Medical Association.

### Decision letter and Author response

Decision letter https://doi.org/10.7554/eLife.84375.sa1
Author response https://doi.org/10.7554/eLife.84375.sa2

## Additional files

### Supplementary files
• MDAR checklist

### Data availability
All data is included in the manuscript and generated custom scripts are included on https://github.com/vss11/Label-free-autofluorescence (copy archived at *Shah, 2023*).

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
