## [Editor Report]

The study addresses a significant impediment to deriving maximal value from living tissue culture systems, which is simultaneously distinguishing and mapping the activity of multiple constituent cell types over the course of experimental perturbations. The authors present a label-free approach that involves collecting autofluorescence and morphological features that provide distinct signatures for mouse tracheal epithelium and demonstrate its application for live imaging of secretory cells. This platform may be very valuable for answering specific experimental questions about tracheal cell behavior in disease.

---

## [Decision Letter]

**Decision letter after peer review:**

Thank you for submitting your article "Label free autofluorescence imaging permits comprehensive and simultaneous assignment of cell type identity and reveals the existence of airway secretory cell associated antigen passages (SAPs)" for consideration by *eLife*. Your article has been reviewed by 2 peer reviewers, and the evaluation has been overseen by a Reviewing Editor and Paul Noble as the Senior Editor. The reviewers have opted to remain anonymous.

Essential revisions:

Both reviewers find great merit in the interesting study. However, they both raise important issues that we would like the authors to address below:

A) Reproducibility:

– Figure 3E, list how many cells were measured in each instance.

– Figure 3F unclear from figure or legend what dimension reduction algorithm was used (PCA? Other?).

– Figure 3 Suppl 2 is not useful as authors don't analyze subsets, any data can form new clusters based on the method used.

– Code should ideally be shared in Github, or in a containerized format in supplement, or anything other than in a PDF file.

B) Minimalism:

– Can the same be achieved just with morphological characteristics? (w/o FAD/NAD(P)H). Using your already existing data, repeat classification analysis omitting the ratios, and show the effect on accuracy/clustering.

– Alternatively, figure 3E: Show which features contribute most to detection in each case. E.g.: Secretory cells' low ratio, ciliated vs ionocyte YZ aspect, etc.

C) Generalizability:

– Describe in conclusions the conditions in which you predict that the method may need to be re-optimized (e.g. inflammation, infection, others) due to metabolic changes.

– Change P3L83: "autofluorescence signature *under the conditions studied in this work* can be used.."

– Relatedly, using your already existing data, is your ability to detect cell types maintained after Rotenone, Antimycin A, or FCCP? this would provide a clue as to the limitations. This loss of resolution is alluded to in P23L328 although not clear how much this is affected.

D) SAPs:

– Using your already existing data, quantify what proportion of secretory cells show the "SAP-like" empty structures at baseline.

– Figure 5C put times in figure (1, 9, 31, 40, 53, 60 minutes).

– Supplemental video: add a timescale.

– The methacholine time course should be improved by quantification of more instances of secretion, and by including appropriate controls:

– Compare SAP formation in absence of methacholine stimulation. Epithelia are very sensitive.

– How often do secretory cells take up and secrete FITC-dextran? This analysis does not need to be extensive as you are only claiming the "existence" of the SAPs, however, would be best to show more than one time only.

E) Others:

– Cite the preprint by coauthor Vinarsky (Kwok et al. 2022) when describing ex vivo prep.

– Cite Kretschmer et al. 2016 (airway-immune interaction, did not separate cell types).

– P2L50, unclear what authors refer to, reference Lin et al. is listed as "submitted" and not available, either remove mention or update citation.

– P23L321, "333% decrease in the CCSP" clarify, as absolute fluorescence value should not possibly decrease by more than 100%.

– Figure 5 confusing choice of colors as Tomato is shown as the same color as Draq5.

1) For the validation in Figure 2, it's not clear whether the cell type was identified by the autofluorescence and then confirmed by the immunostaining, or if and how "positive" and "negative" cells were inferred by autofluorescence. For instance, in Figure 2A, some cells are relatively bright in yellow fluorescence but negative for acetylated tubulin. If this is because the yellow fluorescence pseudocolor is only one feature of the ratiometric signature, that seems fine, but it would be useful to somehow demonstrate visually that the autofluorescence-based identification is independent of the immunostaining. It's just not clear if this is the case from the legend. Panel iii legend says it is an overlay of the staining and autofluorescence but to my eye, it looks like just the autofluorescence with a dotted line demarcating the cell boundaries.

2) Segmentation of single cells by autofluorescence looks excellent for ciliated and secretory cells but is less obvious for basal cells (Figure 2C). Was this done manually (drawing the dashed line) or is the workflow capable of performing this accurately independent of the immunostaining?

3) In Figure 3, it's difficult to get a sense of both of these issues (mentioned in 1 and 2) since in all of the panels A-D it is not easy to distinguish the "positive" cells or to segment the cell by eye based on the autofluorescence image shown.

4) The accuracy of cell type discrimination is quite good however it would be helpful to provide more details about how this was defined and performed. What was the "ground truth" for the cell type used to measure the accuracy? Did this involve immunostaining for cell-type markers?

5) It would be helpful to know how long mouse tracheal explants can be cultured with reasonable maintenance of the autofluorescence features.

---

## [Author Response]

A) Reproducibility:– Figure 3E, list how many cells were measured in each instance.

The numbers of each cell type measured are now included in the Figure 3 legend.

– Figure 3F unclear from figure or legend what dimension reduction algorithm was used (PCA? Other?).

We have added that UMAP was used for clustering in the Figure 3 legend.

– Figure 3 Suppl 2 is not useful as authors don't analyze subsets, any data can form new clusters based on the method used.

We included Figure 3 Suppl 2 purely to facilitate visualization of the clusters as those with red/green color blindness may have difficulty perceiving similar colors.

– Code should ideally be shared in Github, or in a containerized format in supplement, or anything other than in a PDF file.

The code has been uploaded to GitHub under: https://github.com/vss11/Label-free-autofluorescence.

B) Minimalism:– Can the same be achieved just with morphological characteristics? (w/o FAD/NAD(P)H). Using your already existing data, repeat classification analysis omitting the ratios, and show the effect on accuracy/clustering.

We appreciate this suggestion as it further strengthens the rationale and approach for this methodology. We have added Figure 3 Supplement 4 where we perform modeling without autofluorescence data, which shows a dramatic reduction in accuracy with a Matthew’s correlation coefficient ranging from 0.66 to 0.78.

– Alternatively, figure 3E: Show which features contribute most to detection in each case. E.g.: Secretory cells' low ratio, ciliated vs ionocyte YZ aspect, etc.

In addition to the above, we have also now provided the relative contribution of each characteristic to the cell type identification by the Xgboost algorithm in Table 1.

C) Generalizability:– Describe in conclusions the conditions in which you predict that the method may need to be re-optimized (e.g. inflammation, infection, others) due to metabolic changes.

We agree that this is an exceptionally important point for discussion and future directions. We have included a new paragraph with this discussion in the text:

“Despite its advantages, it should be noted that infection and inflammation may alter autofluorescence (Dilipkumar et al., 2019). Thus, it is likely that our methodology must be optimized in each new experimental setting and that machine learning algorithms will require re-training for deployment in disease states.”

– Change P3L83: "autofluorescence signature under the conditions studied in this work can be used.."

We appreciate the suggestion and have made the addition to the text.

– Relatedly, using your already existing data, is your ability to detect cell types maintained after Rotenone, Antimycin A, or FCCP? this would provide a clue as to the limitations. This loss of resolution is alluded to in P23L328 although not clear how much this is affected.

As suggested by the Reviewer, we re-analyzed the data after Antimycin A + Rotenone and FCCP to determine if the autofluorescence ratio is sufficiently different to identify ciliated and secretory cells and included this data in Figure 2 Supplement 1. Though the ratio is affected, it is still useful for cell type identification after intervention, as the ciliated and secretory cells have statistically different ratios. We comment on this further in the Discussion.

D) SAPs:– Using your already existing data, quantify what proportion of secretory cells show the "SAP-like" empty structures at baseline.

We have performed further analysis on Figure 4C as per the reviewer suggestion and found that SAP-like structures are present in ~9% of secretory cells prior to stimulation and dramatically increase to 78% after methacholine stimulation. This finding has been added to the text and Figure 4 Source Data 1.

– Figure 5C put times in figure (1, 9, 31, 40, 53, 60 minutes).

The annotations of the time points are now included in Figure 5D.

– Supplemental video: add a timescale.

The video now has a time scale.

– The methacholine time course should be improved by quantification of more instances of secretion, and by including appropriate controls:

More instances have been included with additional controls (Figure 5C, Figure 5 Supplement 1, Figure 5 Videos 1-3).

– Compare SAP formation in absence of methacholine stimulation. Epithelia are very sensitive.

As per the reviewer’s suggestion, we have now compared SAP formation in unstimulated and methacholine stimulated conditions as seen in Figure 5C. We performed additional analysis of Figure 4C and determined that SAP formation does occur at baseline prior to stimulation in 9% of secretory cells. Methacholine leads to 78% of secretory cells to form SAPs in this image (Figure 4 Source Data 1).

– How often do secretory cells take up and secrete FITC-dextran? This analysis does not need to be extensive as you are only claiming the "existence" of the SAPs, however, would be best to show more than one time only.

We have quantified SAP formation from Figure 4C as discussed above.

We now include 3 videos with instances of uptake and secretion of FITC-Dextran. We have also included a cross section of air-liquid interface culture to show that this process does occur frequently (Figure 5 Supplement 1).

E) Others:– Cite the preprint by coauthor Vinarsky (Kwok et al. 2022) when describing ex vivo prep.– Cite Kretschmer et al. 2016 (airway-immune interaction, did not separate cell types).

We appreciate that the reviewers bringing this study to our attention. We now include it in the manuscript in the introduction:

“And immune cell interactions have also been seen using autofluorescence in the murine trachea (Kretschmer et al., 2016).”

– P2L50, unclear what authors refer to, reference Lin et al. is listed as "submitted" and not available, either remove mention or update citation.

This reference has been removed.

– P23L321, "333% decrease in the CCSP" clarify, as absolute fluorescence value should not possibly decrease by more than 100%.

We appreciate this input and have revised the text to indicate a 70% decrease in fluorescence compared to the time zero sample.

– Figure 5 confusing choice of colors as Tomato is shown as the same color as Draq5.

We have changed the color scheme so that Draq5 (nucleus) is visualized as blue instead of magenta.

1) For the validation in Figure 2, it's not clear whether the cell type was identified by the autofluorescence and then confirmed by the immunostaining, or if and how "positive" and "negative" cells were inferred by autofluorescence. For instance, in Figure 2A, some cells are relatively bright in yellow fluorescence but negative for acetylated tubulin. If this is because the yellow fluorescence pseudocolor is only one feature of the ratiometric signature, that seems fine, but it would be useful to somehow demonstrate visually that the autofluorescence-based identification is independent of the immunostaining. It's just not clear if this is the case from the legend. Panel iii legend says it is an overlay of the staining and autofluorescence but to my eye, it looks like just the autofluorescence with a dotted line demarcating the cell boundaries.

As per the Reviewer comments, we have provided further clarity to describe the process of identifying each cell type and then assigning the measured autofluorescence for each cell type:

“We first measured and recorded autofluorescence using live tissue and then subsequently identified cell types in that same tissue following fixation. Specifically, autofluorescence was first imaged in live tissues, locations were registered to subepithelial landmarks, and the same fixed tissue was subsequently stained with conventional markers to identify cell types. Live autofluorescence signatures were then associated with each cell that was identified by conventional staining. In order to register autofluorescence live images to that of fixed and stained images, we imaged the second harmonic generation (SHG) signals of the subepithelial collagen. These signals served as fiduciary marks which facilitated exact localization of individual cells.”

Regarding the yellow fluorescence pseudocoloring, the ciliated cells do have a strong FADH signal seen as very fine yellow fluorescence, however the presence of yellow fluorescence is not limited to ciliated cells. Furthermore, the measured characteristic is the average fluorescence intensity in the entire cell rather than a binary “yellow present” vs “yellow absent”.

We have also further clarified in the Figure 2 legend to indicates that the dotted line demarcates the cell border in Panel iii.

2) Segmentation of single cells by autofluorescence looks excellent for ciliated and secretory cells but is less obvious for basal cells (Figure 2C). Was this done manually (drawing the dashed line) or is the workflow capable of performing this accurately independent of the immunostaining?

This was done manually by comparing the fixed immunostained image to the autofluorescence image. The workflow is not independently able to draw the borders at this time.

3) In Figure 3, it's difficult to get a sense of both of these issues (mentioned in 1 and 2) since in all of the panels A-D it is not easy to distinguish the "positive" cells or to segment the cell by eye based on the autofluorescence image shown.

We hope with the above clarifications, this issue is clearer. As mentioned, the cell types were identified with immunostaining. Autofluorescence was then measured retrospectively from captured lived images of the same sample.

A major conclusion from Figure 3A-D is that autofluorescence alone may not be sufficient to identify the rare cells. Thus, it is expected that the Reviewer may have difficulty identifying the rare cells by this measure in isolation. This was the rationale for Figure 3E in which we show that quantifying non-invasive morphologic measures provides a method to further discriminate the rare cell populations. Of note, the non-invasive morphologic measurements can be determined using autofluorescence alone.

4) The accuracy of cell type discrimination is quite good however it would be helpful to provide more details about how this was defined and performed. What was the "ground truth" for the cell type used to measure the accuracy? Did this involve immunostaining for cell-type markers?

We appreciate this feedback and now provide additional clarity regarding this approach as described above. The “ground truth” was determined after the live imaging using immunostaining. Autofluorescence and other characteristics measured from live imaging were then attributed to each cell type after identification in post-hoc analysis. This has now been clarified in the text.

5) It would be helpful to know how long mouse tracheal explants can be cultured with reasonable maintenance of the autofluorescence features.

As described in Figure 1E tracheal explants can be cultures for at least 2 days with no significant changes in autofluorescence features.